# Decreasing human body temperature in the United States since the Industrial Revolution

**Myroslava Protsiv[1], Catherine Ley[1], Joanna Lankester[2], Trevor Hastie[3,4], Julie Parsonnet[1,5]***

[1]Division of Infectious Diseases and Geographic Medicine, Department of Medicine, Stanford University, School of Medicine, Stanford, United States; [2]Division of Cardiovascular Medicine, Stanford University, School of Medicine, Stanford, United States; [3]Department of Statistics, Stanford University, Stanford, United States; [4]Department of Biomedical Data Science, Stanford University, School of Medicine, Stanford, United States; [5]Division of Epidemiology, Department of Health Research and Policy, Stanford University, School of Medicine, Stanford, United States

**Abstract** In the US, the normal, oral temperature of adults is, on average, lower than the canonical 37°C established in the 19th century. We postulated that body temperature has decreased over time. Using measurements from three cohorts—the Union Army Veterans of the Civil War (N = 23,710; measurement years 1860–1940), the National Health and Nutrition Examination Survey I (N = 15,301; 1971–1975), and the Stanford Translational Research Integrated Database Environment (N = 150,280; 2007–2017)—we determined that mean body temperature in men and women, after adjusting for age, height, weight and, in some models date and time of day, has decreased monotonically by 0.03°C per birth decade. A similar decline within the Union Army cohort as between cohorts, makes measurement error an unlikely explanation. This substantive and continuing shift in body temperature—a marker for metabolic rate—provides a framework for understanding changes in human health and longevity over 157 years.

***For correspondence:**
parsonnt@stanford.edu

**Competing interests:** The authors declare that no competing interests exist.

## Introduction

In 1851, the German physician Carl Reinhold August Wunderlich obtained millions of axillary temperatures from 25,000 patients in Leipzig, thereby establishing the standard for normal human body temperature of 37°C or 98.6 °F (range: 36.2–37.5°C [97.2- 99.5 °F]) (*Mackowiak, 1997*; *Wunderlich and Sequin, 1871*). A compilation of 27 modern studies, however (*Sund-Levander et al., 2002*), reported mean temperature to be uniformly lower than Wunderlich's estimate. Recently, an analysis of more than 35,000 British patients with almost 250,000 temperature measurements, found mean oral temperature to be 36.6°C, confirming this lower value (*Obermeyer et al., 2017*). Remaining unanswered is whether the observed difference between Wunderlich's and modern averages represents true change or bias from either the method of obtaining temperature (axillary by Wunderlich vs. oral today) or the quality of thermometers and their calibration (*Mackowiak, 1997*). Wunderlich obtained his measurements in an era when life expectancy was 38 years and untreated chronic infections such as tuberculosis, syphilis, and periodontitis afflicted large proportions of the population (*Murray et al., 2015*; *Tampa et al., 2014*; *Richmond, 2014*). These infectious diseases and other causes of chronic inflammation may well have influenced the 'normal' body temperature of that era.

The question of whether mean body temperature is changing over time is not merely a matter of idle curiosity. Human body temperature is a crude surrogate for basal metabolic rate which, in turn,

has been linked to both longevity (higher metabolic rate, shorter life span) and body size (lower metabolism, greater body mass). We speculated that the differences observed in temperature between the 19th century and today are real and that the change over time provides important physiologic clues to alterations in human health and longevity since the Industrial Revolution.

## Results

In men, we analyzed: a) 83,900 measurements from the Union Army Veterans of the Civil War cohort (UAVCW) obtained between 1862 and 1930, b) 5998 measurements from the National Health and Nutrition Examination Survey I cohort (NHANES) obtained between 1971 and 1975, and c) 230,261 measurements from the Stanford Translational Research Integrated Database Environment cohort (STRIDE) obtained between 2007 and 2017 (*Table 1*). We also compared temperature measurements in women within the two later time periods (NHANES, 9303 measurements; and STRIDE, 348,006 measurements).

Overall, temperature measurements were significantly higher in the UAVCW cohort than in NHANES, and higher in NHANES than in STRIDE (*Figure 1*; *Figure 1—figure supplement 1*). In each of the three cohorts, and for both men and women, we observed that temperature decreased with age with a similar magnitude of effect (between −0.003°C and −0.0043°C per year of age, *Figure 1*). As has been previously reported (*Eriksson et al., 1985*), temperature was directly related to weight and inversely related to height, although these associations were not statistically significant in the UAVCW cohort. Analysis using body mass index (BMI) and BMI adjusted for height produced similar results (*Figure 1—figure supplement 2*) and analyses including only white and black subjects (*Figure 1—figure supplement 3*) showed similar results to those including subjects of all ethnicities.

In both STRIDE and a one-third subsample of NHANES, we confirmed the known relationship between later hour of the day and higher temperature: temperature increased 0.02°C per hour of the day in STRIDE compared to 0.01°C in NHANES (*Figure 1*, *Figure 1—figure supplement 2*, *Figure 1—figure supplement 4*). The month of the year had a relatively small, though statistically significant, effect on temperature in all three cohorts, but no consistent pattern emerged (*Figure 1—figure supplement 5*). Using approximated ambient temperature for the date and geographic location of the examination in UAVCW and STRIDE, a rise in ambient temperature of one degree Celsius correlated with 0.001 degree (p<0.001) and 0.0004 degree (p=0.013) increases in body temperature in UAVCW and STRIDE, respectively. Because the seasonal and climatic effects were small and the independent variables were unavailable for many measurements, we omitted month and estimated ambient temperature from further models.

We explored whether chronic infectious diseases—even in the absence of a diagnosis of fever—might raise temperature in the UAVCW cohort, by assessing the temperatures of men reporting a history of malaria (N = 2,203), syphilis (N = 465), or hepatitis (N = 24), or with active tuberculosis (N = 738), pneumonia (N = 277) or cystitis (N = 1,301). Only those currently diagnosed with tuberculosis or pneumonia had elevated temperatures compared to the remainder of the UAVCW population [37.22°C (95% CI: 37.20–37.24°C) and 37.06°C (95% CI: 37.03–37.09°C), respectively compared to 37.02 (95% CI: 36.52–37.53)] (*Supplementary file 1*).

One possible reason for the lower temperature estimates today than in the past is the difference in thermometers or methods of obtaining temperature. To minimize these biases, we examined changes in body temperature by birth decade within each cohort under the assumption that the method of thermometry would not be biased on birth year. Within the UAVCW, we observed a significant birth cohort effect, with temperatures in earlier birth decades consistently higher than those in later cohorts (*Figure 2*). With each birth decade, temperature decreased by −0.02°C. We then assessed change in temperature over the 197 birth-year span covered by the three cohorts. We observed a steady decrease in body temperature by birth cohort for both men (−0.59°C between birth decades from 1800 to 1997; −0.030°C per decade) and women (−0.32°C between 1890 and 1997; −0.029°C per decade). Black and white men and women demonstrated similar trends over time (*Figure 3*).

**Table 1.** Demographic characteristics (N (%)) and mean (SD)) of cohort participants included in the analyses.

| | Total, N (%) | Uavcw | Nhanes i | Stride |
|---|---|---|---|---|
| Individuals | 189,338 (100%) | 23,710 (13%) | 15,301 (8%) | 150,280 (79%) |
| Observations[1] | 677,423 (100%) | 83,699 (12%) | 15,301 (2%) | 578,222 (85%) |
| **Age (years)** | | | | |
| Overall* | | 56.89 (8.85) | 46.55 (16.74) | 53.00 (15.62) |
| 20–40 | 144,379 (21%) | 1682 (2%) | 6489 (42%) | 136,181 (24%) |
| 40–60 | 283,059 (42%) | 52,117 (62%) | 4422 (29%) | 225,365 (39%) |
| 60–80 | 249,985 (37%) | 28,900 (35%) | 4390 (29%) | 216,676 (37%) |
| Weight (Kg) | 0 (0%) | 0 (0%) | 0 (0%) | 0 (0%) |
| Overall* | | 68.63 (10.54) | 70.44 (15.76) | 78.53 (19.75) |
| >60 | 123,931 (18%) | 16,147 (19%) | 4245 (28%) | 103,516 (18%) |
| 60–80 | 296,244 (44%) | 57,475 (69%) | 7311 (48%) | 231,312 (40%) |
| 80–100 | 175,598 (26%) | 9054 (11%) | 3115 (20%) | 163,402 (28%) |
| >100 | 81,650 (12%) | 1023 (1%) | 630 (4%) | 79,992 (14% |
| **Height (cm)** | | | | |
| Overall* | | 172.34 (6.8) | 166.31 (9.17) | 167.78 (10.46) |
| <160 | 145,964 (64%) | 2587 (3%) | 4077 (27%) | 139,295 (24%) |
| 160–180 | 432,404 (64%) | 69,506 (83%) | 9995 (65%) | 352,762 (61%) |
| 180–200 | 98,320 (15%) | 11,569 (14%) | 1227 (8%) | 85,470 (15%) |
| >200 | 735 (0%) | 37 (0%) | 2 (0%) | 695 (0%) |
| **Sex** | | | | |
| Women[2] | 357,309 (53%) | 0 (0%) | 9303 (61%) | 348,006 (60%) |
| Men | 320,114 (47%) | 83,699 (100%) | 5998 (39%) | 230,216 (40%) |
| **Ethnicity** | | | | |
| Black | 68,955 (10%) | 20,801 (25%) | 2399 (16%) | 45,689 (8%) |
| White | 381,330 (56%) | 62,898 (75%) | 12,716 (83%) | 305,581 (53%) |
| Other | 78,277 (12%) | 0 (0%) | 186 (1%) | 78,091 (14%) |
| Unknown | 148,861 (22%) | 0 (0%) | 0 (0%) | 148,861 (26%) |

SD: standard deviation; UAVCW: Union Army Veterans of the Civil War; NHANES: National Health and Nutrition Examination Survey I; STRIDE: Stanford Translational Research Integrated Database Environment; BMI: body mass index. * Mean (SD). [1] Between one and four temperature measurements were available per person. [2]UAVCW included men only.

The online version of this article includes the following source data for Table 1:

**Source code 1.** R code for *Table 1*.

## Discussion

In this study, we analyzed 677,423 human body temperature measurements from three different cohort populations spanning 157 years of measurement and 197 birth years. We found that men born in the early 19[th] century had temperatures 0.59°C higher than men today, with a monotonic decrease of −0.03°C per birth decade. Temperature has also decreased in women by −0.32°C since the 1890s with a similar rate of decline (−0.029°C per birth decade). Although one might posit that the differences among cohorts reflect systematic measurement bias due to the varied thermometers and methods used to obtain temperatures, we believe this explanation to be unlikely. We observed similar temporal change within the UAVCW cohort—in which measurement were presumably obtained irrespective of the subject's birth decade—as we did between cohorts. Additionally, we

saw a comparable magnitude of difference in temperature between two modern cohorts using thermometers that would be expected to be similarly calibrated. Moreover, biases introduced by the method of thermometry (axillary presumed in a subset of UAVCW vs. oral for other cohorts) would tend to underestimate change over time since axillary values typically average one degree Celsius lower than oral temperatures (*Sund-Levander et al., 2002*; *Niven et al., 2015*). Thus, we believe the observed drop in temperature reflects physiologic differences rather than measurement bias. Other findings in our study—for example increased temperature at younger ages, in women, with increased body mass and with later time of day—support a wealth of other studies dating back to the time of Wunderlich (*Wunderlich and Sequin, 1871*; *Waalen and Buxbaum, 2011*).

Resting metabolic rate is the largest component of a typical modern human's energy expenditure, comprising around 65% of daily energy expenditure for a sedentary individual (*Heymsfield et al.,*

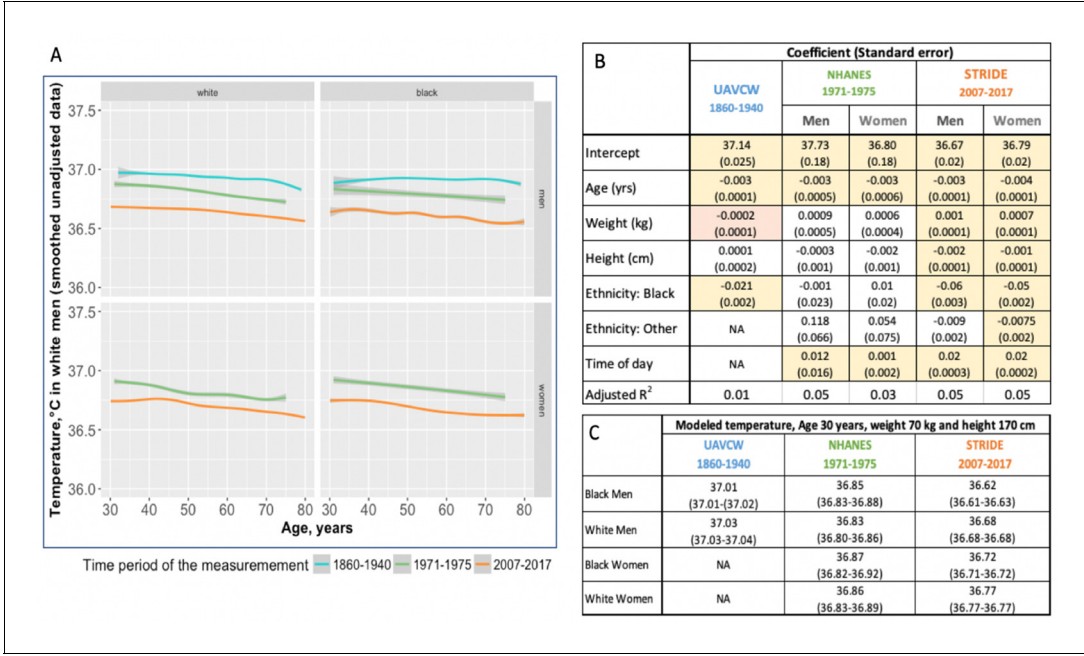

**Figure 1.** Body temperature measurements by age as observed in three different time periods: 1860–1940 (UAVCW), 1971–1975 (NHANES 1), and 2007–2017 (STRIDE). (**A**) Unadjusted data (local regression) for temperature measurements, showing a decrease in temperature across age in white men, black men, white women, and black women, in the three cohorts. (**B**) Coefficients and standard errors from multivariate linear regression models for each cohort including age, weight, height, ethnicity group and time of day as available. Yellow cells are statistically significant at a p value of < 0.01, orange cells are of borderline significance (p<0.1 but>0.05), and remaining uncolored cells are not statistically significant. (**C**) Expected body temperature for 30 year old men and women with weight 70 kg and height 170 cm in each time period/cohort.

The online version of this article includes the following source data and figure supplement(s) for figure 1:

**Source code 1.** R code for *Figure 1*.

**Figure supplement 1.** Distribution of temperature measures (°F) for each cohort: UAVCW (1860–1940), NHANES I (1971–1975) and STRIDE (2007–2017).

**Figure supplement 1—source code 1.** R code for *Figure 1—figure supplement 1*.

**Figure supplement 2.** Analysis using BMI adjusted for height.

**Figure supplement 2—source code 1.** R code for *Figure 1—figure supplement 2*.

**Figure supplement 3.** Model of mean body temperature in black and white ethnicity groups in three different time periods (cohorts): 1860–1940 (UAVCW), 1971–1975 (NHANES 1), and 2007–2017 (STRIDE).

**Figure supplement 3—source code 1.** R code for *Figure 1—figure supplement 3*.

**Figure supplement 4.** Model of mean body temperature over time in three cohorts by birth year controlling for the time of the day of the temperature measurement (white men and women).

**Figure supplement 4—source code 1.** R code for *Figure 1—figure supplement 4*.

**Figure supplement 5.** Effect of the month of temperature measurement.

**Figure supplement 5—source code 1.** R code for *Figure 1—figure supplement 5*.

**Figure supplement 6.** Analysis using both month and location.

**Figure supplement 6—source code 1.** R code for *Figure 1—figure supplement 6*.

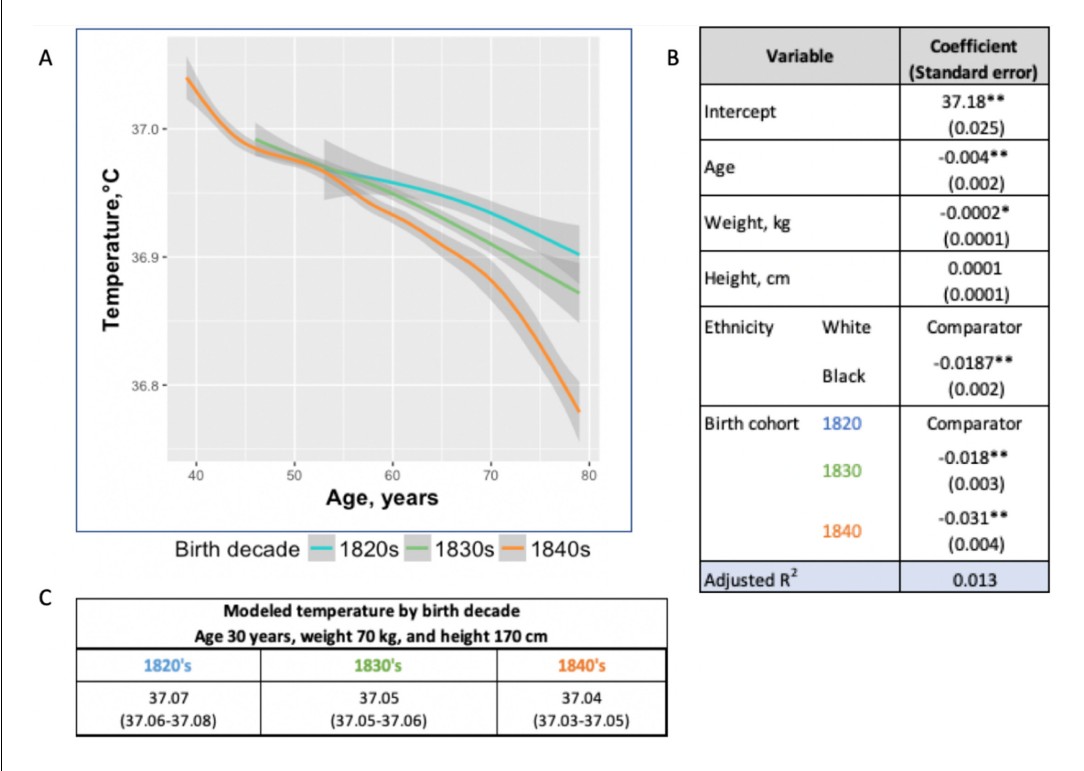

**Figure 2.** Temperature trends within birth cohorts of the UAVCW, 1860–1940 (black and white men). (**A**) Smoothed unadjusted data (local regression) for temperature measurement trends within birth cohorts. The different colors represent different birth cohorts (green: 1820s, blue: 1830s, orange: 1840s). (**B**) Coefficients (and standard errors) from multivariate linear regression including age, body weight, height and decade of birth (1820–1840) (these coefficients do not correspond to the graph as here the trajectories are approximated by linear functions). Only the three birth cohorts with more than 8000 members are included. * and ** indicate significance at the 90%, and 99% level, respectively. (**C**) Expected body temperature (and associated 95% confidence interval) for 30 year old men with body weight 70 kg and height 170 cm in each birth cohort. These values derive from the regression models presented in B.

The online version of this article includes the following source code for figure 2:

**Source code 1.** R code for *Figure 2*.

*2006*). Heat is a byproduct of metabolic processes, the reason nearly all warm-blooded animals have temperatures within a narrow range despite drastic differences in environmental conditions. Over several decades, studies examining whether metabolism is related to body surface area or body weight (***Du Bois, 1936***; ***Kleiber, 1972***), ultimately, converged on weight-dependent models (***Mifflin et al., 1990***; ***Schofield, 1985***; ***Nelson et al., 1992***). Since US residents have increased in mass since the mid-19[th] century, we should have correspondingly expected increased body temperature. Thus, we interpret our finding of a decrease in body temperature as indicative of a decrease in metabolic rate independent of changes in anthropometrics. A decline in metabolic rate in recent years is supported in the literature when comparing modern experimental data to those from 1919 (***Frankenfield et al., 2005***).

Although there are many factors that influence resting metabolic rate, change in the population-level of inflammation seems the most plausible explanation for the observed decrease in temperature over time. Economic development, improved standards of living and sanitation, decreased chronic infections from war injuries, improved dental hygiene, the waning of tuberculosis and malaria infections, and the dawn of the antibiotic age together are likely to have decreased chronic inflammation since the 19[th] century. For example, in the mid-19[th] century, 2–3% of the population would have been living with active tuberculosis (***Tiemersma et al., 2011***). This figure is consistent with the UAVCW Surgeons' Certificates that reported 737 cases of active tuberculosis among 23,757 subjects (3.1%). That UAVCW veterans who reported either current tuberculosis or pneumonia had a higher temperature (0.19°C and 0.03°C respectively) than those without infectious conditions supports this

theory (**Supplementary file 1**). Although we would have liked to have compared our modern results to those from a location with a continued high risk of chronic infection, we could identify no such database that included temperature measurements. However, a small study of healthy volunteers from Pakistan—a country with a continued high incidence of tuberculosis and other chronic infections—confirms temperatures more closely approximating the values reported by Wunderlich (mean, median and mode, respectively, of 36.89°C, 36.94°C, and 37°C) (**Adhi et al., 2008**).

Reduction in inflammation may also explain the continued drop in temperature observed between the two more modern cohorts: NHANES and STRIDE. Although many chronic infections had been conquered before the NHANES study, some—periodontitis as one example (**Capilouto and Douglass, 1988**)— continued to decrease over this short period. Moreover, the use of anti-inflammatory drugs including aspirin (**Luepker et al., 2015**), statins (**Salami et al., 2017**) and non-steroidal anti-inflammatory drugs (NSAIDs) (**Lamont and Dias, 2008**) increased over this interval, potentially reducing inflammation. NSAIDs have been specifically linked to blunting of body temperature, even in normal volunteers (**Murphy et al., 1996**). In support of declining inflammation in the modern era, a study of NHANES participants demonstrated a 5% decrease in abnormal C-reactive protein levels between 1999 and 2010 (**Ong et al., 2013**).

Changes in ambient temperature may also explain some of the observed change in body temperature over time. Maintaining constant body temperature despite fluctuations in ambient temperature consumes up to 50–70% of daily energy intake (**Levine, 2007**). Resting metabolic rate (RMR), for which body temperature is a crude proxy, increases when the ambient temperature decreases below or rises above the thermoneutral zone, that is the temperature of the environment at which humans can maintain normal temperature with minimum energy expenditure (**Erikson et al., 1956**). In the 19th century, homes in the US were irregularly and inconsistently heated and never cooled. By the 1920s, however, heating systems reached a broad segment of the population with mean nighttime temperature continuing to increase even in the modern era (**Mavrogianni et al., 2013**). Air conditioning is now found in more than 85% of US homes (**US Energy Information Administration, 2011**). Thus, the amount of time the population has spent at thermoneutral zones has markedly increased, potentially causing a decrease in RMR, and, by analogy, body temperature.

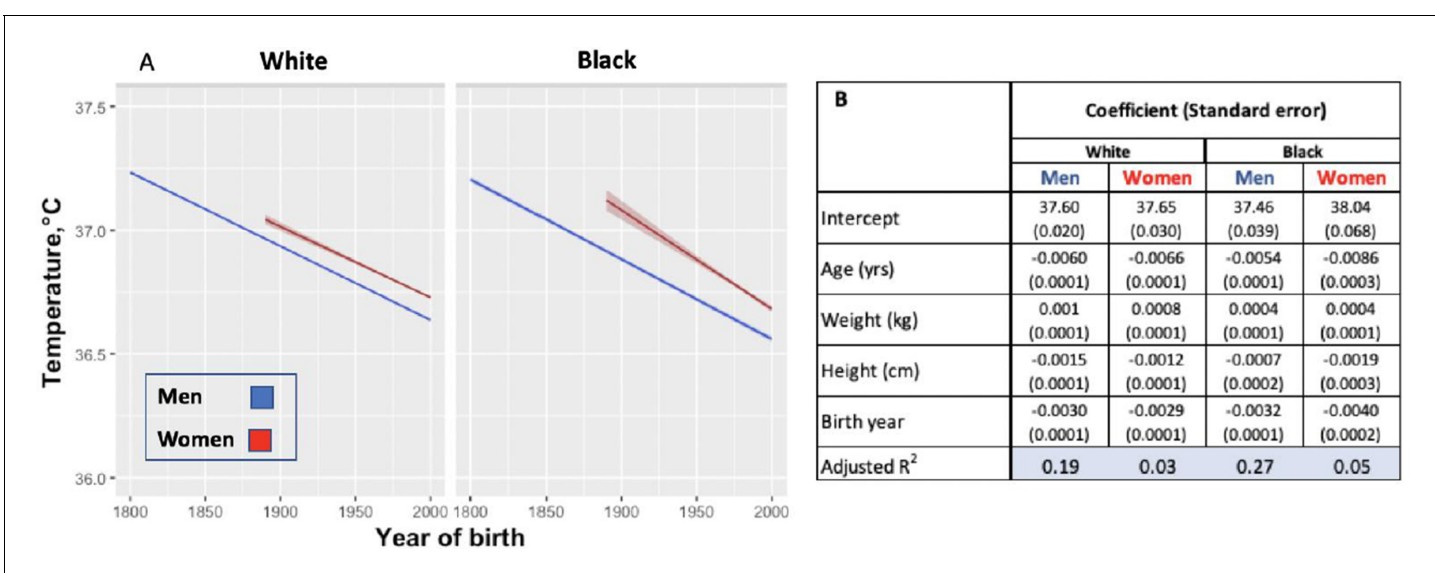

**Figure 3.** Modeled body temperature over time in three cohorts by birth year (black and white ethnicity groups). (**A**) Body temperature decreases by birth year in white and black men and women. No data for women were available for the birth years from 1800 to 1890. (**B**) Coefficients (and standard errors) used for the graph from multivariate linear regression including age, body weight, height and birth year. All cells are significant at greater than 99% significance level.
The online version of this article includes the following source code for figure 3:

**Source code 1.** R code for **Figure 3**.

Some factors known to influence body temperature were not included in our final model due to missing data (ambient temperature and time of day) or complete lack of information (dew point) (*Obermeyer et al., 2017*). Adjusting for ambient temperature, however, would likely have amplified the changes over time due to lack of heating and cooling in the earlier cohorts. Time of day at which measurement was conducted had a more significant effect on temperature (*Figure 1—figure supplement 4*). Based on the distribution of times of day for temperature measurement available to us in STRIDE and NHANES, we estimate that even in the worst case scenario, that is the UAVCW measurements were all were obtained late in the afternoon, adjustment for time of day would have only a small influence (<0.05˚C) on the −0.59˚C change over time.

In summary, normal body temperature is assumed by many, including a great preponderance of physicians, to be 37˚C. Those who have shown this value to be too high have concluded that Wunderlich's 19[th] century measurements were simply flawed (*Mackowiak, 1997*; *Sund-Levander et al., 2002*). Our investigation indicates that humans in high-income countries have changed physiologically over the last 200 birth years with a mean body temperature 1.6% lower than in the pre-industrial era. The role that this physiologic 'evolution' plays in human anthropometrics and longevity is unknown.

## Materials and methods

### Cohorts

We compared body temperature measurements from three cohorts. <u>Cohort 1</u>: The Union Army Veterans of the Civil War, 1860–1940 (UAVCW) is a database from the 'Early Indictors of Later Work Levels, Disease and Death Study', initiated by the late Nobel Laureate, Robert Fogel in 1978 (*Fogel and Wimmer, 1992*) and continuing today. The study abstracted the Compiled Military Service Records, the Pension Records, Carded Medical Records, the Surgeons' Certificates (detailed medical records) and information from the US Federal Census for a cluster sample of Union Army companies in the US Civil War. In total, 331 companies of white and 52 companies of black Union Army veterans were included in the dataset. The Surgeons' Certificates were obtained at locations throughout the US for veterans seeking pension benefits. These certificates include comprehensive medical histories and physical examinations. Body temperatures in Fahrenheit were hand-written on 83,900 Surgeons' Certificates from 23,710 individuals (mean: 3.53 examinations per individual; *Table 1*). Whether the temperatures were taken orally or in the axilla is unknown; both methods were employed in the 19[th] century although oral temperature was more common (*Salinger and Kalteyer, 1900*). Precision of the instruments is also unknown. Inspection of the distribution of reads, however, suggest that it is no better than 0.2 degrees Fahrenheit, consistent with the hashmarks on mercury thermometers (*Figure 1—figure supplement 1*). The UAVCW data—including birth date, temperature, height, weight, location and date of the medical visit, medical history, ongoing medical complaints and findings of physical examinations —are freely available on-line in digital format (The Colored Troops (USCT) original and expanded datasets; *Fogel et al., 2000*; *Costa, 2019*). <u>Cohort 2</u>: The National Health and Nutrition Examination Survey (NHANES I) is a multistage, national probability survey conducted between 1971 and 1975 in the US civilian population. A subset of subjects, aged 1 to 74 years (N = 23,710) underwent a medical examination (ICPSR study No. 8055), including 15,301 adults. The major focus of NHANES I was nutrition, and persons with low income, pregnant women and the elderly were consequently oversampled (*Centers for Disease Control, National Center for Health Statistics, 1975*). Data abstracted included weight, height, sex, ethnicity, and month and geographic region of examination and, as available, time of day the temperature was obtained. In NHANES, mercury thermometers were used and temperatures were taken orally. Precision, as with the UAVCW cohort, is assumed to be 0.2 ˚F. The medical examination was performed by a physician with the help of a nurse. <u>Cohort 3</u>: The Stanford Translational Research Integrated Database Environment (STRIDE) extracts electronic medical record information from patient encounters at Stanford Health Care (Stanford, CA). All adult outpatient encounters at Stanford Health Care from 2007 to 2017 with recorded temperature measurements in the electronic medical record are included in this study (N = 578,522 adult outpatient encounters). Temperature measurements were obtained orally with annually-calibrated, digital thermometers with precision of 0.1 ˚F and extracted from the dataset along with age, sex, weight, height, primary concern at the visit, prescribed

medications, other conditions in the health record with ICD10 codes, and year and time of day the temperature was obtained (mean: 3.85 examinations per individual; *Table 1*).

For the UAVCW and STRIDE datasets, any observations having a diagnosis of fever at the time of the medical examination were excluded. From all three datasets, any extreme values of temperature (<35°C and >39°C) were also excluded from the analysis either because they were implausible or because they indicated a diagnosis of fever and would otherwise have been excluded. Improbable values of both body weight (<30 kg and >200 kg) and height (<120 cm and >220 cm) were also removed. In the UAVCW, we also excluded veterans born after 1850, because they were unlikely to have served in the Union Army.

The use of the STRIDE data was approved as an expedited protocol by the Stanford Institutional Review Board (protocol 40539) and informed consent was waived since the only personal health information abstracted was month of clinic visit. Anonymized data from NHANES and the data from UAVCW are freely available on-line for research use.

## Data analysis

Ethnicity categories were defined differently across cohorts. UAVCW included only white and black men. For comparability, we restricted analyses between the UAVCW and other cohorts to men in these two ethnicity groups. Asians were categorized as 'Other' in NHANES and as 'Asian' in STRIDE, so were considered as 'Other ethnicity' in combined analyses. We performed analyses stratified by sex to account for known temperature differences between men and women. The NHANES study uses sample weights to account for its design; these were incorporated into models including NHANES data (*Centers for Disease Control, National Center for Health Statistics, 1975*).

To estimate the average body temperature during each of the three time periods, we modeled temperature within each cohort using multivariate linear regression, simultaneously assessing the effects of age, body weight, and height. Measurements in men and women were analyzed separately, by white and black ethnicity groups. We also conducted mixed effects modeling to account for the repeated temperature measurements from some individuals. Because the coefficients were almost identical to those of the linear regression models, we chose to present this more simple statistical method. We also assessed the effects of geography, that is location at which temperature was obtained, on temperature (*Figure 1—figure supplement 6*).

To evaluate temperature changes over time, we predicted body temperature using multivariate linear regression including age, body weight, height and birth decade in the UAVCW cohort (the timeframe of NHANES and STRIDE spanned relatively few years, with insufficient variability to evaluate birth cohort effects within these datasets). To assess change in temperature over the 197 birth-year span covered by the three cohorts (between years 1800 and 1997 for men, and between 1890 and 1997 for women), we used linear regression with temperature as the outcome and age, weight, height, and birth decade as independent variables, stratifying by ethnicity and sex. The UAVCW cohort was further investigated for reported infectious conditions that might affect temperature. Diagnoses of infectious conditions, either in the medical history (malaria, syphilis, hepatitis) or active at the time of examination (tuberculosis, pneumonia or cystitis), were included in regression models if fever was not listed as part of that record.

Some models included time of day, ambient temperature and month of year. Time of day at which temperature was taken was available for STRIDE and a subset of NHANES. For individuals without time of day, we imputed the time to be 12:00 PM (noon). We accounted for ambient temperature using the date and geographic location of examination (available in UAVCW and STRIDE) based on data from the National Centers for Environmental Information (*NOAA National Centers for Environmental Information, 2018*). We used the month of year when each measurement was taken as a random effect. To assess the robustness of our result to the chosen methodology, we repeated the analyses using linear mixed effect modeling, adjusting for multiple measurements.

Within the UAVCW, minimum ages varied across birth cohorts due to the bias inherent in the cohort structure (for example, it is impossible to be younger than 30 years of age at the time of the pension visit, be born in 1820s, and be a veteran of the Civil War). To avoid instability in the analysis due to having too few people within specific age groups per birth decade, we excluded the lowest 1% of observations in each birth cohort according to age.

All analyses were performed using R statistical software version 3.3.0. and packages easyGgplot2, lme4, merTools, and ggplot2 for statistical analysis and graphs (www.r-project.org).

## Acknowledgements

We thank Professor Dora Costa, University of California, Los Angeles and Dr Louis Nguyen, Harvard Medical School for sharing their expertise and knowledge of the Union Army data. Thank you also to Dr. Philip Mackowiak, University of Maryland, for providing feedback on study design, analysis and interpretation. We also thank Michelle Bass, PhD, and Yelena Nazarenko for their support of the STRIDE clinical databases.

## Additional information

### Funding

| Funder | Grant reference number | Author |
| --- | --- | --- |
| Stanford Center for Clinical and Translational Research and Education | SPECTRUM award | Julie Parsonnet |

The funders had no role in study design, data collection and interpretation, or the decision to submit the work for publication.

### Author contributions

Myroslava Protsiv, Data curation, Software, Formal analysis, Validation, Visualization, Methodology, Writing—original draft; Catherine Ley, Formal analysis, Methodology, Writing—review and editing; Joanna Lankester, Methodology, Writing—review and editing; Trevor Hastie, Formal analysis, Visualization, Methodology, Writing—review and editing; Julie Parsonnet, Conceptualization, Resources, Supervision, Funding acquisition, Investigation, Visualization, Methodology, Project administration, Writing—review and editing

### Author ORCIDs

Myroslava Protsiv https://orcid.org/0000-0002-7787-5898
Catherine Ley https://orcid.org/0000-0002-8424-7873
Joanna Lankester http://orcid.org/0000-0003-0709-6722
Julie Parsonnet https://orcid.org/0000-0001-7342-5366

### Ethics

Human subjects: The use of the STRIDE data was approved as an expedited protocol by the Stanford Institutional Review Board (protocol 40539) and informed consent was waived since the only personal health information abstracted was month of clinic visit. Anonymized data from NHANEs and the data from UAVCW are freely available on-line for research use.

### Decision letter and Author response

Decision letter https://doi.org/10.7554/eLife.49555.sa1
Author response https://doi.org/10.7554/eLife.49555.sa2

## Additional files

### Supplementary files

- Source data 1. Cohort datasets for all analyses.
- Source code 1. R code - Data import files.
- Source code 2. R code - Data analysis files.
- Source code 3. R code for *Supplementary file 1*.
- Supplementary file 1. Predicted body temperature in individuals with infectious diseases in the UAVCW cohort (1860–1940). Expected body temperature (and associated 95% confidence interval) for a 30 year old white man with body weight of 70 kg and height 170 cm in the UAVCW cohort. *, **, *** indicates significance at the 90%, 95%, and 99% level, respectively.

- Transparent reporting form

## Data availability

All data generated or analysed during this study are included in the manuscript and supporting files.

The following previously published datasets were used:

| Author(s) | Year | Dataset title | Dataset URL | Database and Identifier |
|---|---|---|---|---|
| Fogel RW, Costa DL | 2018 | Aging of Veterans of the Union Army: Surgeons' Certificates, United States, 1862-1940 (ICPSR 2877) | https://doi.org/10.3886/ICPSR02877.v2 | National Archive of Computerized Data on Aging , 10.3886/ICPSR02877.v2 |
| United States Department of Health and Human Services, National Center for Health Statistics | 1992 | National Health and Nutrition Examination Survey I, 1971-1975: Medical Examination (ICPSR 8055) | https://doi.org/10.3886/ICPSR08055.v2 | National Archive of Computerized Data on Aging , 10.3886/ICPSR08055.v2 |

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
