## [Decision Letter]

Thank you for submitting your article "Decreasing human body temperature in the United States since the Industrial Revolution" for consideration by *eLife*. Your article has been reviewed by two peer reviewers, and the evaluation has been overseen by Mark Jit as the Reviewing Editor and Eduardo Franco as the Senior Editor. The following individuals involved in review of your submission have agreed to reveal their identity: Jill Waalen (Reviewer #1); Frank Rühli (Reviewer #2).

The reviewers have discussed the reviews with one another and the Reviewing Editor has drafted this decision to help you prepare a revised submission. We believe that the manuscript is basically acceptable for publication, but would like you to address some minor textual comments raised by the reviewers.

We find that this is an interesting and well-written manuscript using measurements of mean body temperature among 3 US cohorts with data collected in 3 distinct time periods to support the hypothesis that mean population body temperature has decreased over the time from 1860 to 2017. We highly appreciate this work due to its relevant yet unique research topic, solid data sets and straight-forward analysis.

Given the differences in measurement practices, data collection, and environmental conditions over that time period, the paper has tackled a question that is inherently unprovable in regard to cause-effect. However, the arguments are cogent enough (e.g., same known temperature trends corresponding to age seen in all cohorts) and the data supportive enough (e.g., same known temperature trends corresponding to age seen in all cohorts), that it is worth its thought-provocation.

We would like the following textual changes to be made to the manuscript:

1) The last sentence of the Abstract states that understanding the trend of decreasing mean population body temperature "provides a framework for understanding changes in body habitus and human longevity over the last 200 years." There are two problems with this:

a) The study spans 1860-2017, which is 157 years.

b) Given that body temperature is defined as a biomarker for metabolic rate, discussion regarding body habitus in different parts of the manuscript are a bit difficult to reconcile: 1) lower metabolism is associated with greater body mass (Introduction, last paragraph); 2) increased body mass is associated with increased body temperature (Discussion, second paragraph).

Given the above, please change the line in the Abstract to: "provides a framework for understanding changes in human health and longevity over the last 150 years" (or… since the Civil War….)

2) Given the relative paucity of published medical research on the UAVCW cohort (there appears to be only a few PubMed listings), it would be helpful to give more detail about the medical data included for this cohort and how it was obtained:

a) From the brief description in the Materials and methods section, it is not clear how many times an individual may have been examined, for example. Were they tested on a regular basis for pension benefits and disqualified based on some findings? It is stated in the first paragraph of the subsection “Cohorts”, that measures could have been obtained on multiple occasions from individual veterans, which, of course, introduces that problem of lack of independence of measures, but need to have some idea of how big a problem this would be (not likely to be big, but would be good to have numbers).

b) Also, were the examinations done at one location for all veterans (relevant to correlating body temperature with ambient temperature)? This is implied in the third paragraph of the Results, but would be helpful to state explicitly in the Materials and methods.

c) A "Table 1" with basic characteristics of the three cohorts would also be helpful in this regard, given the likely differences, for example, in ranges of height, weight, age, and other covariates. This would be helpful, for example, in understanding why height and weight were not statistically significantly associated in the UAVCW (Results, second paragraph) – perhaps a narrower range in the other cohorts?

3) It is implied in the third paragraph of the subsection “Data Analysis”, that fever was recorded and that these records were excluded in at least some of the models. This raises the question of whether "febrile" subjects were excluded in all cohorts and, if so, what cut-offs were used to define fever in the different cohorts?

4) Given that the changes in mean temperature of interest are relatively small (less than 1 degree), more information regarding the measurement is of interest:

a) In what scale were the measurements in each cohort originally made (Celsius or Fahrenheit)? To what precision level were they recorded?

b) Is it known how thermometers were calibrated for the measures in the respective cohorts (particularly UAVCW)?

5) Please use the term "ethnicity" rather than "race" since we assume this is what is actually being recorded from the cohorts.

6) Discussion, first paragraph: "that" should be "than"

---

## [Author Response]

We believe that the manuscript is basically acceptable for publication, but would like you to address some minor textual comments raised by the reviewers.We find that this is an interesting and well-written manuscript using measurements of mean body temperature among 3 US cohorts with data collected in 3 distinct time periods to support the hypothesis that mean population body temperature has decreased over the time from 1860 to 2017. We highly appreciate this work due to its relevant yet unique research topic, solid data sets and straight-forward analysis.Given the differences in measurement practices, data collection, and environmental conditions over that time period, the paper has tackled a question that is inherently unprovable in regard to cause-effect. However, the arguments are cogent enough (e.g., same known temperature trends corresponding to age seen in all cohorts) and the data supportive enough (e.g., same known temperature trends corresponding to age seen in all cohorts), that it is worth its thought-provocation,We would like the following textual changes to be made to the manuscript:1) The last sentence of the Abstract states that understanding the trend of decreasing mean population body temperature "provides a framework for understanding changes in body habitus and human longevity over the last 200 years." There are two problems with this:a) The study spans 1860-2017, which is 157 years.b) Given that body temperature is defined as a biomarker for metabolic rate, discussion regarding body habitus in different parts of the manuscript are a bit difficult to reconcile: 1) lower metabolism is associated with greater body mass (Introduction, last paragraph); 2) increased body mass is associated with increased body temperature (Discussion, second paragraph).Given the above, please change the line in the Abstract to: "provides a framework for understanding changes in human health and longevity over the last 150 years" (or… since the Civil War….)

We have revised the last sentence of the Abstract to read: “This substantive and continuing shift in body temperature—a marker for metabolic rate—provides a framework for understanding changes in human health and longevity over 157 years”. We feel this version is most understandable to the reader.

2) Given the relative paucity of published medical research on the UAVCW cohort (there appears to be only a few PubMed listings), it would be helpful to give more detail about the medical data included for this cohort and how it was obtained:

We have provided more information on this cohort in the Materials and methods section and added two citations, including one for UAdata.org which provides detailed information and the raw data.

a) From the brief description in the Materials and methods section, it is not clear how many times an individual may have been examined, for example. Were they tested on a regular basis for pension benefits and disqualified based on some findings? It is stated in the first paragraph of the subsection “Cohorts”, that measures could have been obtained on multiple occasions from individual veterans, which, of course, introduces that problem of lack of independence of measures, but need to have some idea of how big a problem this would be (not likely to be big, but would be good to have numbers).

As shown in Table 1, each veteran was tested a mean of 3.5 times (83,699 examinations for 23,710 individuals), often years apart. In STRIDE, temperature measurements were obtained 3.85 times per individual over time. The NHANES subjects’ temperatures were tested just once. We repeated our analyses using mixed effects models to account for repeated measurements and the coefficients were unchanged from the linear regression models. We decided to use the linear regression for simplicity of interpretation. We’ve added a sentence in this regard to the Materials and methods section.

b) Also, were the examinations done at one location for all veterans (relevant to correlating body temperature with ambient temperature)? This is implied in the third paragraph of the Results, but would be helpful to state explicitly in the Materials and methods.

The examinations were conducted at medical examination boards at different locations throughout the country. Veterans may have traveled long distances to have their examinations done. We evaluated whether these locations were associated with variation in temperature since ambient temperature varies considerably throughout the United States. This analysis is now included in the supplement (Figure 1—figure supplement 6). We did not have the location for NHANES (just large regions) and all the STRIDE samples were obtained at our clinics in Northern California. Because of the murkiness of the location data (how long did they reside in these areas, for example, and lack of detailed ambient temperature data for each site), we elected not to include this information in the overall analysis.

c) A "Table 1" with basic characteristics of the three cohorts would also be helpful in this regard, given the likely differences, for example, in ranges of height, weight, age, and other covariates. This would be helpful, for example, in understanding why height and weight were not statistically significantly associated in the UAVCW (Results, second paragraph) – perhaps a narrower range in the other cohorts?

Thank you for this suggestion. We have added means and standard deviations to Table 1 so the reader can better interpret the data.

3) It is implied in the third paragraph of the subsection “Data Analysis, that fever was recorded and that these records were excluded in at least some of the models. This raises the question of whether "febrile" subjects were excluded in all cohorts and, if so, what cut-offs were used to define fever in the different cohorts?

From all three datasets, any values greater than 39ºC were excluded from the analysis because they indicated a diagnosis of fever (subsection “Cohorts”, second paragraph). Temperatures less than 35ºC were also excluded as being implausible. In addition, in both the UAVCW and STRIDE databases, anyone with a physician diagnosis of fever at that visit was excluded from analysis (see the aforementioned paragraph); we excluded them irrespective of the temperature measurement at the visit. Because NHANES did not collect acute illness diagnoses (subjects were volunteers coming to participate in the Nutrition survey and could reschedule for illness), similar information is not available for that cohort.

4) Given that the changes in mean temperature of interest are relatively small (less than 1 degree), more information regarding the measurement is of interest:a) In what scale were the measurements in each cohort originally made (Celsius or Fahrenheit)? To what precision level were they recorded?

The UAVCW Surgeons' Certificates were recorded in Fahrenheit. The precision level is variable (some report to one decimal, others do not) and a histogram of temperature measurements suggests poor precision; temperatures disproportionately fall at a relatively small number of values across the range. The NHANES were collected with mercury thermometers. Because these have hashmarks at 0.2 degree intervals, a disproportionate number of values were “even”. Thus, they should be considered to have 0.2 degree precision as seen in the histogram. We have included Figure 1—figure supplement 1, that presents the distribution of temperature in each of the three cohorts. Reassuringly, inspection of these curves supports the change in mean temperatures over time. Lack of precision, of course, tends to result in negative findings. However, we cannot exclude the possibility that physicians in the UAVCW cohort reported biased values based on prior knowledge of “normal temperature”. Yet this bias would fail to explain the cohort effect within the UAVCW cohort.

b) Is it known how thermometers were calibrated for the measures in the respective cohorts (particularly UAVCW)?

It is not known how or whether thermometers were calibrated for UAVCW or NHANES. For STRIDE, thermometers are calibrated annually by Clinical Technology and Biomedical Engineering within the Stanford Hospitals and Clinics.

5) Please use the term "ethnicity" rather than "race" since we assume this is what is actually being recorded from the cohorts.

Done.

6) Discussion, first paragraph: "that" should be "than"

Done.